# Kinetic and Thermodynamic Characteristics of Fluoride Ions Adsorption from Solution onto the Aluminum Oxide Nanolayer of a Bacterial Cellulose-Based Composite Material

**DOI:** 10.3390/polym13193421

**Published:** 2021-10-05

**Authors:** Alexander V. Dolganov, Vadim D. Revin, Sergey G. Kostryukov, Viktor V. Revin, Guang Yang

**Affiliations:** 1Department of Inorganic and Analytical Chemistry, National Research Ogarev Mordovia State University, 430005 Saransk, Russia; dolganov_sasha@mail.ru (A.V.D.); Vadim.revin.16@mail.ru (V.D.R.); kostryukov-sg@mail.ru (S.G.K.); 2Department of Biotechnology, Bioengineering and Biochemistry, National Research Ogarev Mordovia State University, 430005 Saransk, Russia; 3Department of Biomedical Engineering, Huazhong University of Science and Technology, Wuhan 430074, China; yang_sunny@yahoo.com

**Keywords:** bacterial cellulose, aluminum oxide, fluorine, solution, ALD technology, adsorption

## Abstract

The described research examined the adsorption of fluoride ions from solution immobilized onto an aluminum oxide-coated bacterial cellulose-based composite material in which aluminum oxide had been deposited using ALD technology. The kinetic regularities of the adsorption of fluoride ions from the solution as well as the mechanism of the processes were analyzed. The established equations show that the dynamics of adsorption correspond to first-order kinetics. Based on the Langmuir adsorption isotherms, we defined the adsorption equilibrium constants, parameter maximum adsorption, and change in Gibbs free energy. It is shown that, with increasing temperature, an increase in the reaction rate is constant, both forward and reverse. This testifies to the activated character of adsorption of the first fluoride on the surface of the sorbent based on bacterial cellulose modified with an alumina nanolayer. The activation energy of the desorption process is higher than the activation energy of the adsorption process, which characterizes the adsorption as ionic. The negative value of entropy indicates that in the course of sorption, an adsorption complex “aluminum-fluorine” is formed, where the system is more ordered than the initial system in which fluorine ions are in solution. The limiting stages of the process are revealed. The high sorption capacity of the resulting bacterial cellulose-based composite material obtained by means of biosynthesis through cultivation of the bacterium *Komagataeibacter sucrofermentans* B-11267 was demonstrated.

## 1. Introduction

Fluoride is known to have both a beneficial and harmful effect on human health. Fluoride concentration in between 0.4 and 1.0 mg L^−1^ has a beneficial effect, as it protects us from tooth decay by promoting calcification of dental enamel [1]. However, fluoride concentrations in excess of 1.5 mg L^−1^ leads to serious health problem such as fluorosis, osteoporosis, arthritis, brittle bones, cancer, infertility, brain damage, Alzheimer syndrome, and thyroid disorder [1,2]. In recent times, fluoride contamination of drinking water has emerged as a serious environmental health hazard worldwide with countries such as the United States of America, China, India, Kenya, etc. reporting high concentrations of fluoride in drinking water [2,3]. Thus, it is important to remove excess fluoride from drinking water to maintain the optimal concentration of fluoride. Fluoride in water generally originates due to slow dissociation of fluoride based minerals such as fluorite (CaF2), fluorapatite (Ca5(PO4)3F), and cryolite (Na3AlF6), along with discharge of wastes from industries such as semiconductor manufacturing, coal power plants, electroplating, rubber and fertilizer production [2,4]. Over the years, many treatment methods such as electrodialysis [5], ion exchange [6], adsorption [7,8,9], chemical precipitation, and coagulation [10] have been employed to remove excess fluoride from water. Among the different treatment methods available for removal of the contaminant from water, adsorption has emerged as the most popular and promising technology because of the ease of operation, viability, simplicity of equipment, high efficiency, and low cost of the adsorbents [2,7,11]. In recent years, various adsorbents, such as activated and organic acid-modified alumina [9,12], graphene [8], Fe_3_O_4_ nanocomposites [13,14,15] and Ce-doped bone char [16] have been developed and tested for the removal of fluoride from water. Although a handful of reports are available on the removal of fluoride by different adsorbents, defluoridation of water is still considered to be an active field of research and hence needs the attention of the researchers to develop inexpensive but efficient adsorbents for fluoride removal. The removal of fluoride ions via an adsorption method is considered the most effective, safe, and cost-effective among known methods [16,17]. The adsorption of ions from solution is used in many industries, primarily for cleaning various solutions by removing pollutants.

Currently, natural biopolymers or composites based on them are used as sorbents. In recent years, the demand for biodegradable materials based on plant cellulose has increased. Such cellulose is the most common organic biopolymer worldwide due to its accessibility. Of greater significance, however, are the special physical and chemical properties of bacterial cellulose. A large group of bacteria are able to synthesize cellulose. Among the known strains that we obtained for this study is Komagataeibacter sucrofermentans B-11267 [18,19,20,21,22,23,24,25,26].

The production of bacterial cellulose was a complex and costly process. However, considering modern technologies and the development of processes over time, today the production of bacterial cellulose has become easier, which simplifies its availability. The combination of bacterial cellulose properties dominates cheap cellulosic materials such as waste cellulose fibers from the pulp and paper and textile industries [25].

Bacterial cellulose has a high degree of crystallinity and high polymerization, a large surface area, and a high water retention capacity. These properties allow it to be effectively used in filtering facilities for water defluorination. At a practical level, aluminum oxide can be applied as an active adsorbent, as it has a low cost and a porous structure with the presence of nanoscale cavities. Aluminum oxide is also characterized by low toxicity, which is most important for application in drinking water [27]. In our opinion, combining the best qualities of bacterial cellulose and aluminum oxide in one material will allow obtaining a synergistic effect, which will be reflected in obtaining a material with high values of sorption capacity. Therefore, the aim of this work was to obtain a more efficient bio-based adsorbent to fluoride ions. Bacterial cellulose (BC) is currently considered a functional biomaterial with numerous applications in various fields, including biomedicine for tissue engineering [21,22,23], wound dressing [17,23,24,25,26], and controlled drug delivery systems [23]. BC can be used in dietetics as a carrier of additives for balanced nutrition and in industrial electronics for the manufacture of acoustic diaphragms. It is a promising source for obtaining biocomposite materials [17,26]. BC meets many criteria to fit the profile of a highly efficient adsorbent [28]. The greatest advantage of modifying adsorbents with nanocellulose or nanocellulose-based adsorbents is the number of functional groups they provide due to the high surface area and functional group density. The numerous hydroxyl groups (or others, if chemically modified) on the nanocellulose result in a higher capacity of the target molecule to attach to the adsorbent [24]. Unlike other inorganic adsorbents, nanocellulose-based materials are also totally biodegradable, enabling both sustainability and biological use without side effects.

Herein, we report the synthesis a 50 nm thick nanofilm of aluminum oxide [24], deposited on the surface of bacterial cellulose by means of ALD technology as a sorbent. The synthesized sorbent was then used as an adsorbent for removal of fluoride from water. The isotherm and kinetics of the adsorption process was thoroughly investigated along with its possible mechanism of fluoride uptake.

## 2. Experimental Section

Bacterial cellulose obtained from *Komagataeibacter sucrofermentans* B-11267 was used as an adsorbent. The bacterium was cultured at a temperature of 28 °C for 3 days on slant agar medium (pH 5.0–6.0) with the following composition (in g/L): glucose—10.0; yeast extract—10.0; peptone—7.0; agar—15,0; citric acid—0.2; acetic acid—0.1; ethanol—10.0. This solution was sterilized at 121 °C for 15 min without ethanol in an MLS-3781L autoclave (Sanyo, Japan).

Purification and quantification of bacterial cellulose.

The resulting BC was treated three times with 0.1 N NaOH solution at 80 °C for 30 min to remove cells and medium components. The cellulose was washed from the alkali solution with a 0.5% solution of acetic acid and distilled water until neutral. The amount of the polysaccharide was determined by the gravimetric method, by drying to constant weight at a temperature of 60 °C on a balance of accuracy class II [29].

To obtain the inoculum, we used HS medium for growth, which had the following composition (in g/L): glucose—20.0; peptone—5.0; yeast extract—5.0; sodium hydrophosphate—2.7; citric acid—1.15. The cultures were dispensed in 250 mL flasks containing 100 mL of the medium and grown using the ES-20/60 shaker incubator (BioSan, Latvia) at 250 rpm for 24 h at a temperature of 28 ± 1 °C. The resulting inoculum, 10% of the medium volume, was inoculated in experimental flasks containing 100 mL of medium. Cultivation was performed in a shaker incubator at 250 rpm at a temperature of 28 ± 1 °C for 3–6 days.

Bacterial cellulose was obtained on a liquid medium with molasses (50 g/L final concentration), pH 4.50. The bacterial cellulose was coated with an aluminum oxide nanofilm by means of ALD technology. The process of nanofilm deposition by the atomic layer deposition method is as follows: the substrate, which is located in the vacuum chamber, is exposed to precursors, which are vapors, at the optimal operating temperature. Precursors that react only on the outer surface of the substrate form a thin-film nanolayer of the compound formed during reaction. The thickness of the deposited nanolayers is usually calculated using previously plotted calibration curves [24].

The reactor temperature is usually 100 °C. When depositing the aluminum oxide nanofilm, Al(CH_3_)_3_ (TMA) and H_2_O were used as precursors. The aluminum-containing precursor has good reactivity. Growth occurs owing to self-limiting reactions [24].

To study the sorbent’s kinetic and thermodynamic properties, a simulated fluoride-ions solution concentration equal to 5.0 mg/L was used. This concentration is optimal when studying adsorption processes. A simulated fluoride-ions solution was prepared by diluting the standard sample of fluoride SSS 7261-96 with deionized water. Before studying the physical and chemical processes, the amount of aluminum oxide deposited onto bacterial cellulose was determined in advance. The sample composition was estimated by atomic absorption spectroscopy using a SHIMADZU AA-7000. The Al_2_O_3_ content varied from 42.63 to 68.10% of the total mass of the sorbent, depending on the layer thickness.

The MAS NMR ^27^Al spectra were recorded utilizing the JEOL JNM-ECX400 (9,39 T) spectrometer at a Larmor frequency of 104,23 MHz with the radiofrequency (RF) pulse duration of 0.3 ms. For the NMR experiment, a 3.2 mm diameter zirconium dioxide rotor and a double-resonance sensor were utilized. The rotor frequency was 17 kHz, and the time delay between pulses was 1 s. The total number of scans was 460,000. The spectra were processed using the ACD/NMR Processor Academic Edition, Ver. 12.01.

Samples of bacterial cellulose-based sorbent weighing 10^−2^ g were added to a simulated fluoride-ions solution with a concentration of 5.0 mg/L for 1 h and mixed on a Biosan shaker at a speed of 170 rpm. After adsorption equilibrium was reached, the fluoride-ions content was analyzed by the spectrophotometric method on the SHIMADZU UV-1800 UV–visible spectrophotometer as per GOST 4386-89. Each experiment was performed 3 times under standard conditions. The standard deviation of the measurements was within ±2.0%.

The adsorption capacity with regard to aluminum oxide was calculated using the following Formula (1):(1)$$q=\frac{C_{0}-C}{m}\cdot V$$

The degree of fluoride removal by the resulting sorbent was calculated using the following formula:(2)$$R=\frac{C_{0}-C}{C_{0}}\cdot 100$$

## 3. Results and Discussion

To select the adsorption duration time at which, all other things being equal, it can be assumed that adsorption equilibrium had occurred, we modeled the dependence of the Fluoride ions adsorption value on time (min). The concentration and volume of the adsorptive remained constant throughout the experiment (5.0 mg/L and 10.0 mL, respectively). The resultant dependence of the adsorption value on time is shown in Figure 1.

As shown in Figure 1, the relationship between the adsorption capacity of the sorbent and the reaction time suggests that with an increase in the contact time between the sorbent and the fluoride ion solution, the capacity of the adsorbent based on bacterial cellulose increases. Fluorine adsorption rapidly increased during the first 30 min from the start of the reaction, and then the process slowed down over time. The reaction is completely completed after 60 min and then the system goes into a state of equilibrium. This is explained by the fact that at the initial stage of time, a large number of free active sites were available on the surface of a sorption layer of aluminum oxide 50 nm thick at a concentration of a model fluorine solution of 5.0 mg/L. Over time, it was difficult for fluorine ions to occupy vacant areas on the surface of the sorbent, which slowed down the rate of adsorption. As a result, the reaction was almost complete after 60 min, from which it can be concluded that the optimal time for the sorption of fluoride ions from the solution is 1 h.

Considering the previously selected optimal conditions described in [24], namely, the thickness of the sorbent layer of aluminum oxide is 50 nm, the neutral medium of the solution, the optimal sorption time is 60 min for the adsorption of fluoride ions from solution by the modified sorbent, another experiment was set up to determine the dependence of the adsorption value (A) on different concentrations of fluoride solution (C) (Figure 2). The resulting dependence shown in Figure 2 corresponds to the Langmuir-type and is described using the Langmuir equation. 

The Langmuir equations can be represented as follows:(3)$$A=\frac{A_{\infty }\cdot K\cdot C}{1+K\cdot C}$$
(4)$$\frac{1}{A}=\frac{1}{A_{\infty }}+\frac{1}{A_{\infty }\cdot K\cdot C}$$
where $A_{\infty }$ is the maximum fluoride ions adsorption from solution (mg/g);

*K* is the constant indicating the affinity of the adsorbed substance relative to the surface of the adsorbent.

*C* is the equilibrium concentration of the fluoride-ions solution (mg/L).

Equation (3) corresponds to the standard form, Equation (4) to the linear form.

To calculate the adsorption parameters, we plotted the dependence in the coordinates as *1*/*A* vs. *1*/*C* (Figure 3). The presented isotherm was amenable to linearization, which allowed us to calculate the Langmuir adsorption constants. Using Equation (4) and Figure 3, we calculated the value of *1/A_∞_* and *1*/(*A*_∞_·*K*) [30]. Therefore, the maximum adsorption of fluoride ions *A*_∞_ is 500 ± 9.3 mg/g, and the constant *K* is equal to 1000.

Based on the obtained values of the maximum adsorption and the equilibrium constant, we can describe the dependence of the adsorption value on the concentration of a solution of fluoride ions:(5)$$A=\frac{5\;\times \;10^{5}\cdot C}{1+1000\cdot C}$$

Then, the kinetic parameters were determined for the absorption of fluoride ions in solution by the bacterial cellulose-based modified sorbent. In this regard, adsorption can be characterized as a reversible reaction of the first order [31].
(6)$${\mathrm{AL}}_{2}O_{3}+F^{-}\overset{k_{1}}{\underset{k_{2}}{⇄}}{\left[{\mathrm{AL}}_{2}O_{3}F\right]}^{-}$$
where *k*_1_ is the adsorption rate constant, and *k*_2_ is the desorption rate constant.

Since, in our case, the adsorption process can be described as a first-order equation, the rate of such a reaction can be represented using the following equation:−*dx/dt* = *v*_1_ − *v*_2_ = *k*_1_·(*a* − *x*) − *k*_2_·(*b* + *x*)(7)
where *v*_1_ is the forward reaction rate (on rate);

*v*_2_ is the counter (reverse) reaction rate;

*a* and *b* are some amounts of substance;

*x* is the amount of the substance that has reacted to this point in time *t* [32].

Solving the equation graphically (8) in coordinates $ln\frac{\left(C_{0}-C_{\infty }\right)}{C-C_{\infty }}$ − *t*, we determined the effective constant *k* = *k*_1_
*+ k*_2_: (8)$$k\cdot t=\left(k_{1}+k_{2}\right)\cdot t=\ln\left(C_{0}-\frac{C_{\infty }}{\left(C_{0}-C_{\infty }\right)-(C_{0}-C)}\right)=ln\frac{\left(C_{0}-C_{\infty }\right)}{C-C_{\infty }}$$
where 

$C_{\infty }$ is the concentration of the fluoride-ions solution corresponding to the adsorption equilibrium (mg/L); 

*C* is the concentration of the fluoride-ions solution at the corresponding time *t*, mg/L;

$C_{0}$ is the initial concentration of the fluoride solution, mg/L.

Given that the equilibrium constant *K_c_* is defined as the ratio *k*_1_/*k*_2_, it is also possible to estimate the constants *k*_1_ and *k*_2_ [33]. 

The equilibrium constant *K_c_* was calculated by Equation (9):(9)$$K_{c}=\frac{C_{\infty }}{C_{0}-C_{\infty }}$$

Then, using the Arrhenius Equation (10), we determined the values for the adsorption and desorption activation energies.
(10)$$k=A\cdot e^{\frac{-E_{a}}{RT}}$$
where *E_a_* is the activation energy, J/mol;

*k* is the reaction rate constant;

*T* is the temperature, K;

*A* is the preexponential factor;

*e* is the base of the natural logarithm (Napierian base).

Among the stages of determining the kinetic parameters of sorption was the analysis of changes in fluoride-ions solution concentrations over time at different temperatures, 283, 298, and 310 K, which were regulated using a thermostat. The obtained results are presented in Figure 4 in the form of kinetic curves [34].

The obtained kinetic curves show that the adsorption rate is the highest in the first 30 min, and the system approaches equilibrium by 60 min.

After processing the above findings, using Equation (8), we obtained the logarithmic anamorphoses shown in Figure 5.

In order to plot and then analyze the logarithmic dependence, we considered the precise time period when the reaction was approximately 80% complete. In Figure 4, this section is located in the time interval between 0 and 30 min.

As seen in Figure 5, the logarithmic anamorphoses are linear. This once again confirms that the adsorption of fluoride ions from solution by bacterial cellulose-based modified sorbent is a first order reaction.

The effective (total) reaction constant *k* was calculated from anamorphoses (Figure 5) as a tangent of the angle of the straight-line slope. The resulting values of *k*_1_ and *k*_2_ are shown in Table 1.

As follows from these constants, the rate of forward reaction exceeds that of the reverse. We also observed the increased rate constants of adsorption and desorption with increasing temperature. This determines the activated nature of fluoride ions adsorption from solution immobilized onto the surface of the resulting bacterial cellulose-based sorbent by the aluminum oxide nanolayer.

In order to estimate the activation energy of adsorption and desorption, it is necessary to plot the dependence of the logarithms of the above constants on inverse temperature (1/T). The resulting relationship is shown in Figure 6 below.

The dependence of the adsorption (*k*_1_) and desorption (*k*_2_) rate constants on temperature is effectively described using the Arrhenius equation. The energy of the adsorption and desorption activation (*E_a_*) was estimated by Equation (11) and defined as a tangent of the angle *α* of the straight-line slope:(11)$$tg\alpha =-\frac{E_{a}}{R}$$

The estimated values of the activation energies of adsorption and desorption are 41.761 and 78.572 kJ/mol, respectively. From the obtained findings, we can see that the adsorption process is ionic in nature, which is confirmed by the values of the activation energies of adsorption and desorption.

For a more detailed description of the adsorption of fluoride ions from solution, it was necessary to calculate the following thermodynamic characteristics of these processes: Δ*G*^0^, Δ*H*^0^, and Δ*S*^0^ [35]. Δ*H*^0^ was calculated graphically after reaction-isobar Equation (12) with the coordinates *lnK_c_* vs. 1/T (Figure 7).
(12)$$lnK_{c}=-\frac{\Delta H^{0}}{RT}+C$$

The adsorption enthalpy from Figure 7 was estimated by the following formula:(13)$$\Delta H^{0}=-R\cdot tg\alpha .$$

The change in the Gibbs free energy was determined by the Gibbs–Helmholtz formula [35]:(14)$$\Delta G^{0}=-RT\cdot lnK_{c}$$

After estimation of Δ*G*^0^ и Δ*H*^0^, we calculated the entropy of absorption Δ*S*^0^:(15)$$\Delta S^{0}=\frac{\Delta H^{0}-\Delta G^{0}}{T}$$

The results are presented in Table 2.

From the above data, it follows that the adsorption of fluoride from solution by the bacterial cellulose-based modified sorbent occurs to the fullest extent at low temperatures [36]. In this case, the adsorption process is exothermic. The entropy is less than zero, which implies that during fluoride sorption from solution, the AlF3 complex is formed, where the system is the most ordered. From a thermodynamic point of view, adsorption in this case is a spontaneous process [37]. This is confirmed by the negative Gibbs value. Minor differences in the entropy of fluoride adsorption from solution by the modified sorbent may be indicative of the fact that the adsorption complex has an identical structure [38]. The adsorption of fluorine from the solution by the modified sorbent can be represented as the interaction of the adsorbate molecules with the active centers of the adsorbent surface.

The resulting composite material based on bacterial cellulose modified with an alumina nanolayer using ALD technology has the highest adsorption capacity of fluorine from water (80.4 ± 1.59 mg/g) in comparison with other sorbents, such as pine sawdust, modified with aluminum (up to 4 ± 0.2 mg/g) [18], modified bauxite (10.0 ± 0.3 mg/g) [19], sorbent from mixed oxides Mg-Al synthesized using FCT (55.0 ± 1.1 mg/g) [20]. In work [21], where the declared capacity of the adsorbent Al(OH)_3_ is 0.19 ± 0.004 mg/g. In work [22], the adsorbent is activated aluminum, capacity 1.45 ± 0.03 mg/g); in the study [23], the adsorbent is Al_2_O_3_ impregnated with Mg(OH)_2_, capacity 10.12 ± 0.21 mg/g), the authors of [24] show the adsorbent granular iron hydroxide, capacity 7.0 ± 0.15 mg/g).

To confirm the course of the chemisorption, we studied samples of sorbents before and after sorption by utilizing the energy-dispersive X-ray spectroscopy (EDS) method. The EDS method not only allows information on the chemical composition to be obtained, but also provides a picture of the elemental distribution of the material over the entire sample before and after the sorption of fluoride ions from water (Figure 8).

As seen from the presented figures, the sorption takes its course as chemisorption with the formation of water-insoluble fluorides on the surface.

Aluminum is not washed out of the cellulose surface, which confirms the heterogeneous path of the sorption.

To demonstrate the presence of a nanoscale film of amorphous Al_2_O_3_ on the cellulose surface, as well as the changes that occurred during sorption, solid-state nuclear magnetic resonance with magic angle spinning (MAS NMR ^27^Al) was employed [39].

Figure 9 shows the solid-state NMR ^27^Al spectra of the sorbent samples (1) before and (2) after sorption.

Thus, a broad signal in the region from 80 to −80 ppm corresponds to aluminum nuclei in the Al_2_O_3_ of various polymorphic modifications. After sorption, an additional peak of approximately −17 ppm appeared in the NMR spectrum, corresponding to the aluminum nucleus surrounded by fluorine atoms [39].

## 4. Conclusions

The kinetic regularities of fluoride ions adsorption from solution by a bacterial cellulose-based modified sorbent were studied. It was discovered that, in this case, the adsorption can be described by kinetic equations of the first order. The kinetic and thermodynamic parameters were calculated, which allowed us to reveal the exothermic nature of the above process, effectively occurring at low temperatures. The Gibbs free energy and entropy change had negative values in the studied temperature range. The rate of adsorption is higher than that of desorption, and the activation energy of the desorption process exceeds the activation energy of the adsorption process.

The results confirm the ionic nature of the adsorption. The optimum duration for the process is 60 min, its isotherm is capable of linearization in Langmuir coordinates, and the approximation validity factor is 98.85. Under optimal conditions of sorption, the removal of fluoride ions from the solution occurs by 87.53%, and 12.47% remains. During the course of absorption, it was revealed that the structure of the resulting complex, AlF_3_, remains constant throughout the entire process. This is demonstrated by the negative value of entropy and its minor differences in the specified temperature range.

## Figures and Tables

**Figure 1 polymers-13-03421-f001:**
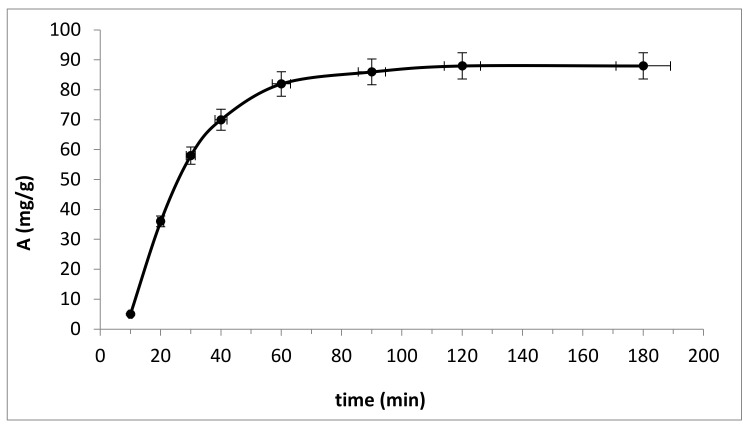
Time course (min) isotherm for fluoride absorption from solution (concentration of fluoride ions 5 mg/L; ratio of adsorbate–adsorbent solution 1000:1 (mL/g)).

**Figure 2 polymers-13-03421-f002:**
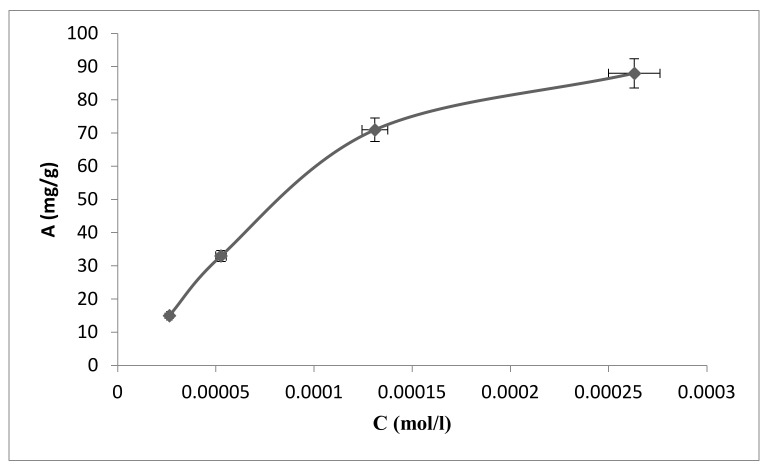
Dependence of the adsorption capacity on the fluoride ions solution concentration (ratio of adsorbate–adsorbent solution 1000:1 (mL/g)).

**Figure 3 polymers-13-03421-f003:**
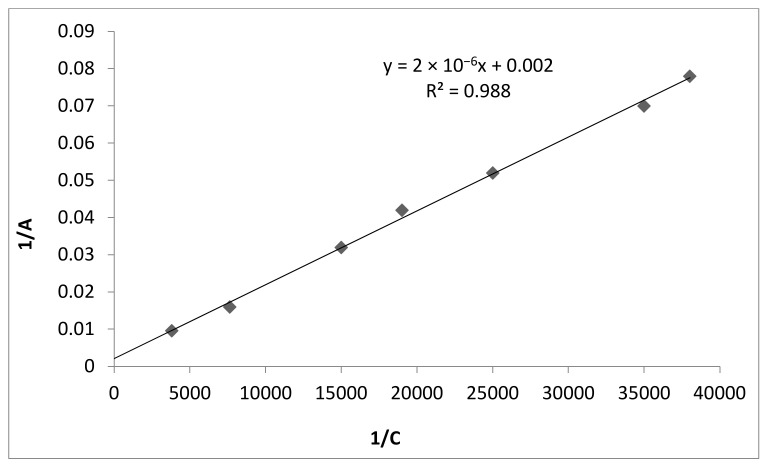
Isotherm for fluoride ions adsorption from solution, represented according to linearized coordinates (ratio of adsorbate-adsorbent solution 1000:1 (mL/g)).

**Figure 4 polymers-13-03421-f004:**
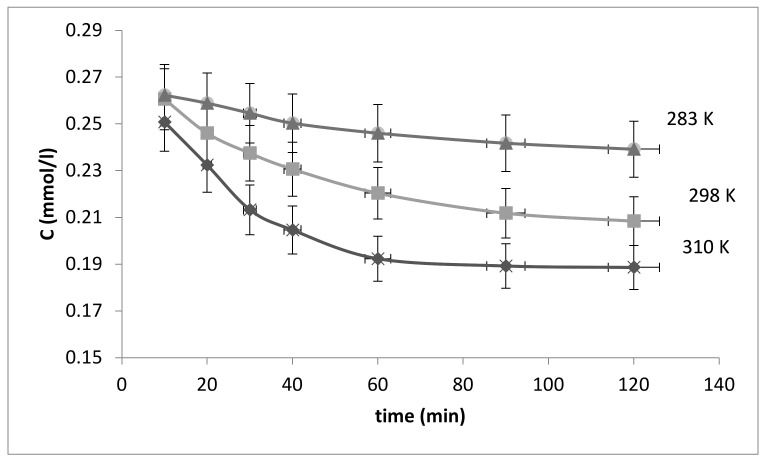
Kinetic curves of fluoride adsorption from solution by the bacterial cellulose-based modified sorbent (ratio of adsorbate–adsorbent solution 1000:1 (mL/g)).

**Figure 5 polymers-13-03421-f005:**
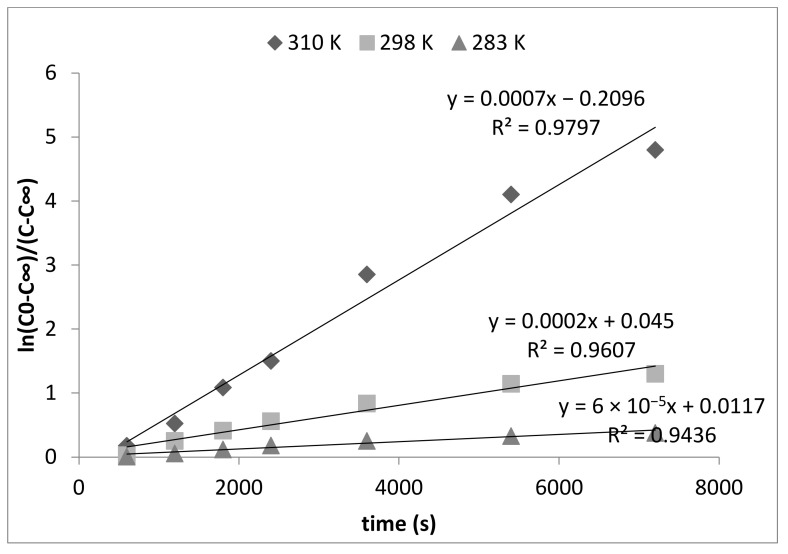
Logarithmic anamorphoses of kinetic curves of fluoride ions absorption from solution by the bacterial cellulose-based modified sorbent.

**Figure 6 polymers-13-03421-f006:**
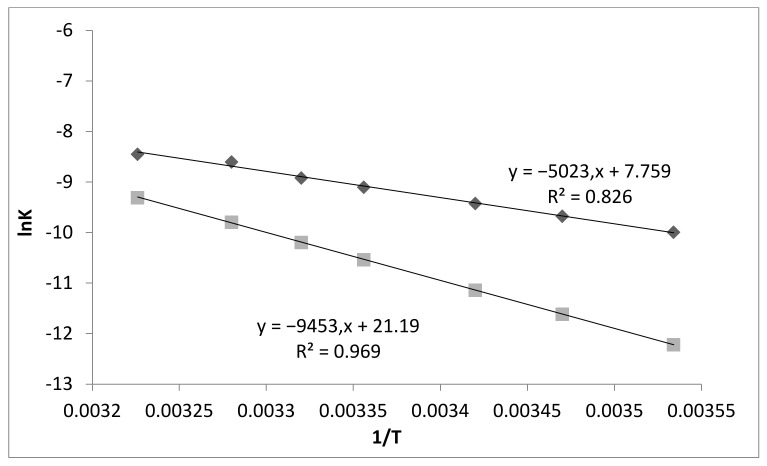
Dependence of the logarithms of the forward and reverse reaction rate constants on inverse temperature.

**Figure 7 polymers-13-03421-f007:**
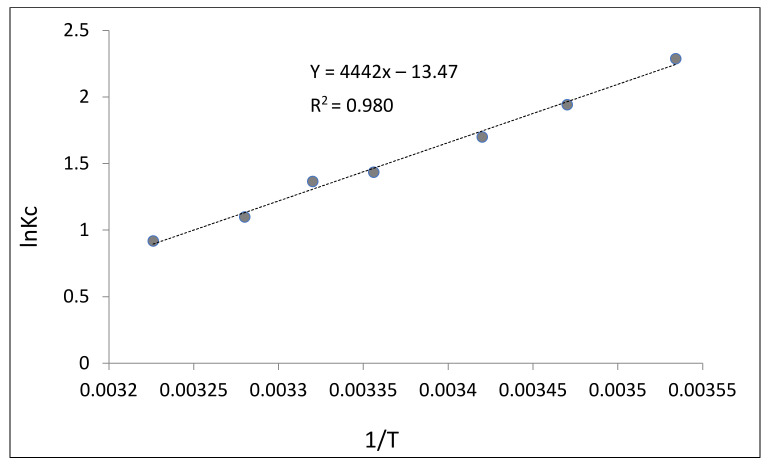
Logarithmic dependence of the equilibrium constants of the fluoride ions adsorption reaction from solution on inverse temperature.

**Figure 8 polymers-13-03421-f008:**
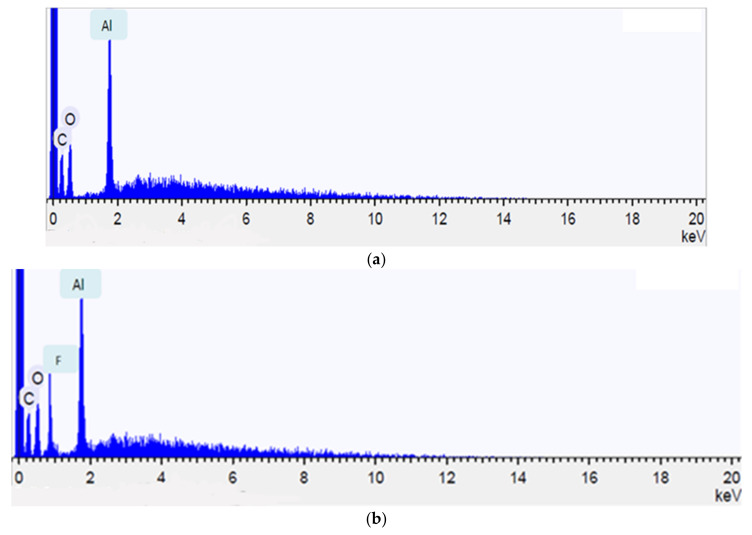
Diagram of the elemental distribution of the sample (**a**) before and (**b**) after sorption, obtained by the EDS method at an accelerating voltage of 15 kV.

**Figure 9 polymers-13-03421-f009:**
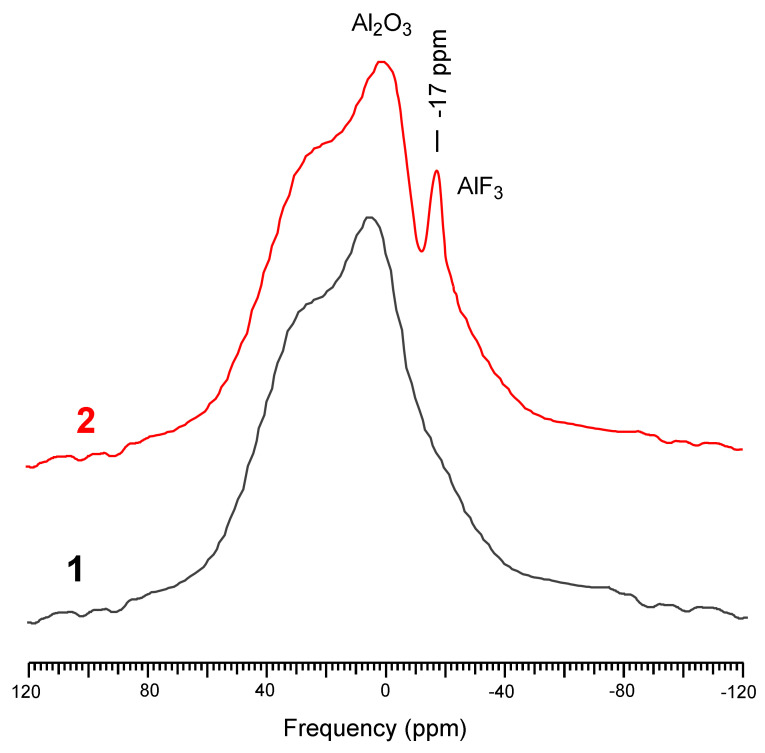
MAS NMR ^27^Al sorbent spectra (1) before and (2) after sorption.

**Table 1 polymers-13-03421-t001:** Kinetic parameters of the adsorption and desorption of fluoride ions from solution by the bacterial cellulose-based modified sorbent at different temperatures.

*T*, *K*	310	298	283
$C_{\infty }$, mol/L	1.88 × 10^−4^	2.08 × 10^−4^	2.38 × 10^−4^
*K_c_* = $C_{\infty }$/(*C*_0_ − $C_{\infty }$)	2.508	3.744	9.672
*k*	3.80 × 10^−4^	9.41 × 10^−5^	6.0 × 10^−5^
*k* _1_	2.72 × 10^−4^	7.43 × 10^−5^	5.44 × 10^−5^
*k* _2_	1.08 × 10^−4^	1.98 × 10^−5^	5.62 × 10^−6^

**Table 2 polymers-13-03421-t002:** Thermodynamic characteristics of fluoride ions sorption from solution by the bacterial cellulose-based modified sorbent.

*T*, *K*	*K_c_*	Δ*H*°, kJ/mol	Δ*G*°, kJ/mol	Δ*S*°, J/molK
298	3.744363	−36.931	−3.271	−112.95

## Data Availability

The data that support the findings of this study are available from the corresponding author upon reasonable request.

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
