# Peer review of "Kinetic and Thermodynamic Characteristics of Fluoride Ions Adsorption from Solution onto the Aluminum Oxide Nanolayer of a Bacterial Cellulose-Based Composite Material"

_polymers, 2021, doi:10.3390/polym13193421_

Round 1
Reviewer 1 Report
(1). Line 201. Remark: Error in coordinates ln.
These coordinates should be ln (see below)
(2). Mathematical error in eq.(7). In reality, ln(Co – ≠ ln
To get equality, this eq. should be corrected, as follows
ln( = ln)
(3). Table 3. Remark: Error in the dimension of ΔSo (J/mol). In reality, the dimension of ΔSo should be J/(mol K)
After correcting these errors, the article can be recommended for publication
Author Response
(1). Line 201. Remark: Error in coordinates ln.
These coordinates should be ln (see below)
In the revised version of article, all recommended changes have been made.
(2). Mathematical error in eq.(7). In reality, ln(Co – ≠ ln
To get equality, this eq. should be corrected, as follows
ln( = ln)
In the revised version of article all recommended changes have been made.
(3). Table 3. Remark: Error in the dimension of ΔSo (J/mol). In reality, the dimension of ΔSo should be J/(mol K)
In the revised version of article all recommended changes have been made.

Reviewer 2 Report
After reading the manuscript entitled: Kinetic and Thermodynamic Characteristics of Fluorideions adsorption from Solution onto the Aluminum Oxide Nanolayer of a Bacterial Cellulose-based Composite Material Prepared Using ALD, which aimed to study the kinetic and themodynamic of adsorption of Flourine ions on the prepared adsorbent.
I regret to say that I cannot recommend the publication of this manuscript in this journal since overall it did not add new, in addition, some of the conclusions made against the experimental results. Furthermore, the following comments are to be considered:
1- Excessive English editing is required.
2- Suitable characterization of the prepared adsorbent must be done such as XRD, SEM, TEM.
3- The abstract and the conclusions contain some misleading and incorrect statements. For example, the equilibrium result obtained does not confirm monolayer adsorption. Also, the data shown in Figure 4 support the endothermic process as mentioned in the abstract while results shown in Table 2 indicates an exothermic process.
4- You cannot use abbreviations in the title, explain what is ADL to the readers. Also on page 2, line 74, what is BC?
5- On page 4, line 155, you stated the formation of 50 nm thick layer of Aluminium oxide. How did you confirm the formation of metal oxide without XRD and how did you measure the thickness?
6- On page 4, line 159, you cannot state that the optimal time is 1h since it is only valid for the experimental conditions used for one trial only. For example, if you change the mass of the adsorbent, the temperature, or the concentration, you will get different equilibrium times.
7-In Figure 2, the experimental points are very low, you need more points to conclude if it is a monolayer or not. This shape is not showing a monolayer.
8- No need for both equations 2 and 3, one of them is enough to show, and usually the linear is preferred.
9- In Figure 4, it is better to show time in minutes rather than seconds.
10- On page 10, line 288, and on page 12, line 344, you stated that the adsorption is exothermic, while Figure 4 shows endothermic nature. The removal efficiency was highest at the highest temperature.
11- On page 10, line 289, you stated the formation of AlF3 complex. How did you confirm this?
12- On page 11 lines 297-305, the comparison based only on adsorption capacity is misleading, you need to consider the similarity of adsorption conditions before comparing the capacity.
13- In Figure 8, the appearance of F peak does not confirm chemical adsorption, physical adsorption will show the peak also.
14- In the conclusion, line 353, how did you confirm structural stability without performing a recycling test?
Author Response
1- Excessive English editing is required.
We polished the English
2- Suitable characterization of the prepared adsorbent must be done such as XRD, SEM, TEM.
The composite was characterized by us earlier doi.org/10.3390/polym13091422
3- The abstract and the conclusions contain some misleading and incorrect statements. For example, the equilibrium result obtained does not confirm monolayer adsorption. Also, the data shown in Figure 4 support the endothermic process as mentioned in the abstract while results shown in Table 2 indicates an exothermic process.
We checked the experimental data again. All findings are consistent with the reported results. In fig. 4 shows the kinetic data, based on them it is not possible to calculate the heat effect of the reaction
4- You cannot use abbreviations in the title, explain what is ADL to the readers. Also on page 2, line 74, what is BC?
We have eliminated these inconsistencies
5- On page 4, line 155, you stated the formation of 50 nm thick layer of Aluminium oxide. How did you confirm the formation of metal oxide without XRD and how did you measure the thickness?
The composite was characterized by us earlier doi.org/10.3390/polym13091422
6- On page 4, line 159, you cannot state that the optimal time is 1h since it is only valid for the experimental conditions used for one trial only. For example, if you change the mass of the adsorbent, the temperature, or the concentration, you will get different equilibrium times.
We agree with the reviewer with this comment. The article cites parameters for the sorption process under experimental conditions.
7-In Figure 2, the experimental points are very low, you need more points to conclude if it is a monolayer or not. This shape is not showing a monolayer.
In the revised version of the article, we have improved the quality of the experimental results.
8- No need for both equations 2 and 3, one of them is enough to show, and usually the linear is preferred.
We have indicated both equations to demonstrate the consistency of our reasoning.
9- In Figure 4, it is better to show time in minutes rather than seconds.
We agree with this and edit fig. 4
10- On page 10, line 288, and on page 12, line 344, you stated that the adsorption is exothermic, while Figure 4 shows endothermic nature. The removal efficiency was highest at the highest temperature.
We calculated the exothermic effect of the reaction without using the data from Fig. 4. We agree that the maximum adsorption is observed with increasing temperature, however, this is not enough to unambiguously indicate the nature of the endothermic effect.
11- On page 10, line 289, you stated the formation of AlF3 complex. How did you confirm this?
The formation of aluminum fluoride is indicated by a signal in the region of -17 ppm. in solid-state 27Al NMR spectrum. This signal was absent in the starting material before sorption. On the spectrum given in the article, everything is indicated
12- On page 11 lines 297-305, the comparison based only on adsorption capacity is misleading, you need to consider the similarity of adsorption conditions before comparing the capacity.
We present the data by comparing the values of the maximum capacity under optimal conditions.
13- In Figure 8, the appearance of F peak does not confirm chemical adsorption, physical adsorption will show the peak also.
Figure 8 shows data indicating the presence of an Al-F bond in the sample. Based on this, it can be concluded that the process has the character of chemical adsorption
14- In the conclusion, line 353, how did you confirm structural stability without performing a recycling test?
In the revised version of article all recommended changes have been made.

Reviewer 3 Report
The issue of water purification is relevant today. The presence of fluorine-containing compounds can cause serious harm to human health. That is why this kind of work deserves attention. I recommend that the editor consider the manuscript for publication, but before that, in my opinion, you should pay attention to the following points:
I recommend the authors add additional points on the dependencies, for example 7, since some of them only have 3 points.
Line 11.I recommend deleting K.
Line 11, 12. It is advisable to give a decoding of the values ​​instead of "A∞, and ΔG0ad".
Lines 53-59. For these statements, it is advisable to add a link, for example, Vinogradov, M.I. et al Rheological Properties of Aqueous Dispersions of Bacterial Cellulose. Processes 2020, 8, 423. https://doi.org/10.3390/pr8040423
Line 87. Why exactly 50 nm? It is necessary to clarify or add a link.
Section 2. What is the yield of bacterial cellulose? How were bacteria residues removed?
Line 142. "minThe" - must be checked.
Line 158. Remove the extra point.
Lines 228, 260, 261, 286 "highestin", "tangentof", "theangle" "bythe" add a space, etc.
Figure 8. Remove all unnecessary information from the graphs.
Lines 326-331. This information should be transferred to the experimental part.
Author Response
The issue of water purification is relevant today. The presence of fluorine-containing compounds can cause serious harm to human health. That is why this kind of work deserves attention. I recommend that the editor consider the manuscript for publication, but before that, in my opinion, you should pay attention to the following points:
I recommend the authors add additional points on the dependencies, for example 7, since some of them only have 3 points.
In the revised version of the article, we have improved the quality of the figures.
Line 11.I recommend deleting K.
In the revised version of article all recommended changes have been made.
Line 11, 12. It is advisable to give a decoding of the values ​​instead of "A∞, and ΔG0ad".
In the revised version of article all recommended changes have been made.
Lines 53-59. For these statements, it is advisable to add a link, for example, Vinogradov, M.I. et al Rheological Properties of Aqueous Dispersions of Bacterial Cellulose. Processes 2020, 8, 423. https://doi.org/10.3390/pr8040423
Thanks for the reference. We used it in the revised version of the article.
Line 87. Why exactly 50 nm? It is necessary to clarify or add a link.
The composite was characterized by us earlier doi.org/10.3390/polym13091422
Section 2. What is the yield of bacterial cellulose? How were bacteria residues removed?
We have inserted in the experimental part data on the purification and quantification of bacterial cellulose.
The resulting BC was treated three times with 0.1 N NaOH solution at 80 ° C for 30 minutes to remove cells and medium components. The cellulose was washed from the alkali solution with a 0.5% solution of acetic acid and distilled water until neutral.The amount of the polysaccharide was determined by the gravimetric method, by drying to constant weight at a temperature of 60 ° C on a balance of accuracy class II
Line 142. "minThe" - must be checked.
In the revised version of article all recommended changes have been made.
Line 158. Remove the extra point.
Lines 228, 260, 261, 286 "highestin", "tangentof", "theangle" "bythe" add a space, etc.
In the revised version of article all recommended changes have been made.
Figure 8. Remove all unnecessary information from the graphs.
In the revised version of this article all recommended changes have been made.
Lines 326-331. This information should be transferred to the experimental part.
In the revised version of article all recommended changes have been made.

Round 2
Reviewer 2 Report
Even though I am against separating the characterization of the adsorbent from the application, I can recommend the publication now.